# Cnidarian-bilaterian comparison reveals the ancestral regulatory logic of the β-catenin dependent axial patterning

Tatiana Lebedeva [1], Andrew J. Aman[1], Thomas Graf[1], Isabell Niedermoser [1], Bob Zimmermann [1], Yulia Kraus[1,2], Magdalena Schatka[1], Adrien Demilly[1], Ulrich Technau [1] & Grigory Genikhovich [1✉]

In animals, body axis patterning is based on the concentration-dependent interpretation of graded morphogen signals, which enables correct positioning of the anatomical structures. The most ancient axis patterning system acting across animal phyla relies on β-catenin signaling, which directs gastrulation, and patterns the main body axis. However, within Bilateria, the patterning logic varies significantly between protostomes and deuterostomes. To deduce the ancestral principles of β-catenin-dependent axial patterning, we investigate the oral–aboral axis patterning in the sea anemone *Nematostella*—a member of the bilaterian sister group Cnidaria. Here we elucidate the regulatory logic by which more orally expressed β-catenin targets repress more aborally expressed β-catenin targets, and progressively restrict the initially global, maternally provided aboral identity. Similar regulatory logic of β-catenin-dependent patterning in *Nematostella* and deuterostomes suggests a common evolutionary origin of these processes and the equivalence of the cnidarian oral–aboral and the bilaterian posterior–anterior body axes.

[1] Department of Neurosciences and Developmental Biology, Faculty of Life Sciences, University of Vienna, Djerassiplatz 1, Vienna, Austria. [2] Department of Evolutionary Biology, Faculty of Biology, Lomonosov Moscow State University, Leninskiye gory 1/12, Moscow, Russia. ✉email: grigory.genikhovich@univie.ac.at

Graded morphogen signals comprise the top tier of the axial patterning cascades in Bilateria and their phylogenetic sister group Cnidaria (corals, sea anemones, jellyfish, hydroids)[1–3]. Just like the posterior–anterior (P–A) body axis of Bilateria, the oral–aboral (O–A) body axis of Cnidaria is patterned by Wnt/β-catenin signaling[4,5] (Fig. 1a). Although it is likely that β-catenin signaling is also involved in the axial patterning of earlier branching ctenophores and sponges[6,7], cnidarians are the earliest branching animal phylum for which experimental gene function analyses are available. A cnidarian–bilaterian comparison can inform us about the ancestral logic of the β-catenin-dependent axial patterning and mechanisms of molecular boundary formation. In this paper, we focus on deciphering the mechanism of the O–A axis patterning in the ectoderm of the early embryo of the sea anemone *Nematostella vectensis*.

Morphologically, the O–A axis in *Nematostella* becomes apparent at the onset of gastrulation, when future endoderm starts to invaginate, eventually forming the inner layer of this diploblastic organism. The establishment of the O–A axis in *Nematostella* depends on β-catenin[8]. Its knockdown abolishes the O–A axis both morphologically and molecularly: the embryos fail to gastrulate and do not express oral ectoderm markers[9]. In contrast, mosaic stabilization of β-catenin results in the formation of numerous ectopic oral structures or even complete ectopic axes[4] (Supplementary Fig. 1a–c). By late gastrula stage, the ectoderm of *Nematostella* can be roughly subdivided into three axial domains: the oral domain characterized by *Brachyury* (*Bra*) expression, the midbody domain where *Wnt2* is expressed, and the aboral domain expressing *Six3/6* (Fig. 1b), whereas endodermal O–A patterning begins later in development[10]. Pharmacological experiments, in which β-catenin signaling was upregulated by a range of concentrations of the GSK3β inhibitor 1-azakenpaullone (AZK) (Fig. 1a), showed that ectodermally expressed β-catenin-dependent genes react to different levels of upregulation of β-catenin signaling dose-dependently and in two distinct ways[4] (Fig. 1c). Some genes, whose expression is normally restricted to the oral ectodermal domain, increase their expression to saturation upon upregulation of β-catenin signaling and start to be expressed in the ectoderm along the whole O–A axis at high AZK concentrations. We call them "saturating" genes below. Other ectodermally expressed genes, whose normal expression can be observed either in the oral domain or further aborally, require permissive "windows" of β-catenin signaling intensities. Upon weak pharmacological upregulation of β-catenin signaling, "window" gene expression shifts aborally, i.e. into the area where endogenous β-catenin signaling intensity is expected to be lower, while upon strong upregulation of β-catenin signaling their expression ceases altogether[4] (Fig. 1c). A similar dose-dependent response to "windows" of β-catenin signaling intensity was previously demonstrated in axial patterning of bilaterians. Particularly striking is the resemblance to the P–A patterning in deuterostomes: the neurectoderm in vertebrates[11,12], body ectoderm in hemichordates[13,14] and sea urchins[15], and endomesoderm in sea stars[15,16]. In protostomes, the P–A axis patterning mechanisms are very diverse, however, the posteriorizing effect of β-catenin signaling can also be observed. Different levels of knockdown of the β-catenin signaling antagonist *Axin* resulted in different extent of posteriorization of the embryo and loss of anterior structures in the short-germ insect *Tribolium castaneum*[17,18]. Conversely, different levels of *Wnt8* knockdown led to the expansion of the anterior and loss of the posterior segments in the spider *Achaearanea tepidariorum*[19]. Within Spiralia, AZK-dependent disappearance of the anterior and expansion of the posterior marker gene expression was observed in the embryos of brachiopods *Novocrania anomala* and *Terebratalia transversa*[20], while experimental up- and downregulation of β-catenin signaling resulted, respectively, in vegetalization and animalization of the embryo of the nemertean *Cerebratulus lacteus*[21], reminiscent of the effect in deuterostomes[14,22,23].

Thus, the regulatory principle behind the "window" behavior may represent the ancestral logic of β-catenin-dependent axial patterning, however, its mechanism is not clear. Since this regulatory behavior is likely to be at the core of the O–A patterning in *Nematostella*, and possibly represents a general mechanism shared by all animals, we attempted to explain it. Since not only oral, but also several midbody and aboral markers were shown to be abolished upon β-catenin knockdown[9], we hypothesized that both "saturating" and "window" genes are positively regulated by β-catenin (Fig. 1a). However, in order to account for the repression of the "window" genes upon upregulation of β-catenin, we postulated that there exists a "transcriptional repressor X", which, being a "saturating" gene, becomes upregulated upon increased β-catenin signaling and inhibits the expression of the "window" genes in ever more aboral positions and, eventually, throughout the embryo (Fig. 1a, c). In this study, we set out to test our assumption and search for this hypothetical repressor. We demonstrate that a unit of four transcription factors, Bra, FoxA, FoxB and Lmx, rather than a single transcriptional repressor X, is responsible for controlling the "window" gene behavior in the oral domain of the *Nematostella* embryo. We also show that the regulatory logic based on repression of the more aborally expressed β-catenin signaling target genes by the more orally expressed β-catenin signaling target genes is responsible for setting up gene expression domain boundaries along the entire O–A axis and identify Sp6-9 as a "transcriptional repressor Y" setting up the midbody/aboral boundary. We argue that this represents the ancestral regulatory logic of β-catenin-dependent axial patterning conserved since before the cnidarian–bilaterian split and discuss the implications of this on our understanding of the correspondence of the cnidarian and bilaterian body axes.

## Results

**Identification of the transcriptional repressor X candidates.** Our hypothesis predicted that: (i) the transcriptional repressor X is to be found among the "saturating" genes upregulated upon increased β-catenin signaling, (ii) it has to be expressed in a contiguous domain along the O–A axis rather than in a salt-and-pepper manner to be able to act cell-autonomously, and that (iii) the loss of function of the transcriptional repressor X will abrogate the β-catenin-dependent repression of "window" genes converting them into "saturating" genes upon pharmacological upregulation of the β-catenin signaling (Fig. 1c). To test these predictions, we devised an RNA-Seq-based strategy for finding all transcription factors fulfilling these criteria (Fig. 1d). In order to obtain an off-target free list of transcription factors upregulated by β-catenin, we used two independent means of upregulating β-catenin signaling by suppressing the activity of two different members of the β-catenin destruction complex, which we further refer to as "treatments". First, we used AZK treatment spanning different time windows to suppress GSK3β. Second, we used a line of *Nematostella* carrying a frameshift mutation in the *APC* gene[24] (Fig. 1a, e). At 3 days post fertilization (3 dpf), all $APC^{-/-}$ embryos display a phenotype similar to that of embryos incubated from early blastula on in AZK (Supplementary Fig. 1d–h). Visual detection of the homozygous *APC* mutants at 1 dpf is impossible, since the phenotype only becomes apparent at 2–3 dpf. However, an earlier study showed that "window" behavior of *Wnt2* persisted until at least 3 dpf[5], which suggested that the putative repressor X was expressed both at 1 dpf and at 3 dpf. Therefore, we compared the transcriptomes of 1 dpf embryos and 3 dpf embryos incubated in AZK with the transcriptomes of the 3

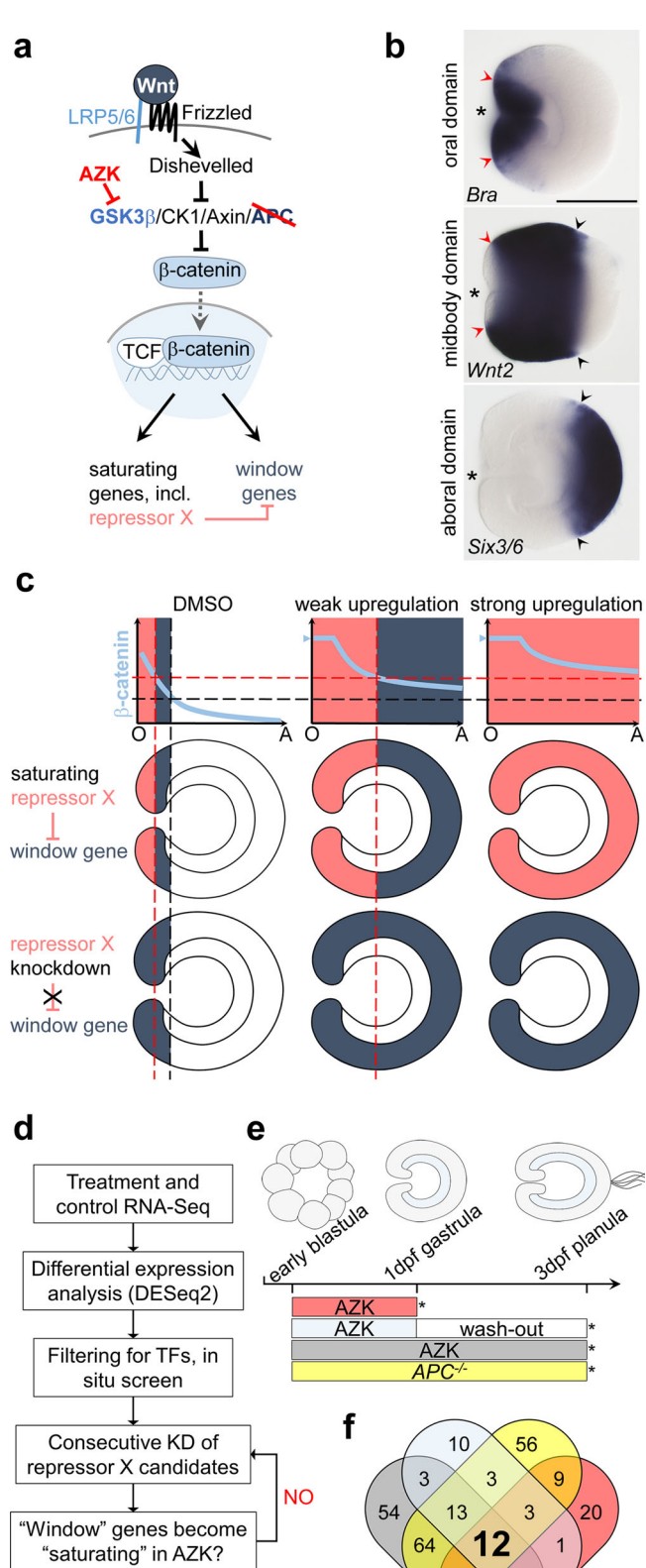

**Fig. 1 The "repressor X" concept and the search strategy. a** Scheme of the Wnt/β-catenin signaling pathway indicating the members manipulated in this study in order to artificially upregulate it. We use two types of treatments (red) to upregulate β-catenin signaling: pharmacological inhibition of GSK3β by AZK and mutation of *APC*. **b** Oral, midbody, and aboral domains of the 1 day post fertilization (1 dpf) gastrula visualized by molecular markers. Lateral views, oral to the left. Asterisk denotes the blastopore. Arrowheads demarcate corresponding positions. Scale bar 100 μm. **c** Hypothetical mechanism of the response of the "saturating" and "window" genes to different intensities of the β-catenin signaling and the putative role of the transcriptional repressor X in regulating the "window" expression behavior. Hypothetical oral-to-aboral gradient of β-catenin signaling is shown in light blue on the upper panels. Repressor X is a saturating gene expressed above a certain β-catenin signaling intensity indicated by the red dashed line, i.e., orally (pink expression domain on graphs and middle panels). The window gene (blue expression domain) is activated above the β-catenin signaling intensity indicated by the black dashed line, however, it becomes repressed in the area of repressor X expression. Upon AZK treatment, the β-catenin signaling intensity increases eventually reaching saturation (blue arrowhead on the Y-axis). In increasing AZK concentrations, the minimal β-catenin signaling intensity sufficient for repressor X activation shifts aborally, displacing the area available for the window gene expression until it becomes impossible for the window gene to be expressed anywhere in the embryo. Upon repressor X knockdown (bottom panel), the window gene starts to behave as a saturating gene. O and A on graphs indicate the oral and the aboral end. **d** Search strategy used to identify transcriptional repressor X. **e** Scheme of treatments. At 1 dpf, AZK treatments were stopped at 30 h post fertilization (hpf), and either RNA was extracted immediately, or the embryos were washed out and incubated in *Nematostella* medium until 3 dpf (72 hpf). Asterisks indicate time points of RNA extraction. **f** Venn diagram with the numbers of the putative transcription factor coding genes upregulated by different treatments. The color code corresponds to that on **e**.

Supplementary Table 1) of which we excluded five: two as metabolic enzymes falsely annotated by INTERPROSCAN[25] as transcription factors (*NVE21786* and *NVE12602*), one, *MsxC*, since it was not expressed in the wild type gastrula, and two, *Unc4* and *AshC*, because they were expressed in single cells rather than in contiguous domains (Supplementary Fig. 2g). The remaining seven candidates, *Brachyury (Bra)*, *FoxA*, *FoxB*, *LIM homeobox (Lmx)*, *Shavenbaby (Svb)*, *Dachshund (Dac)* and a putative Zn finger transcription factor *NVE11868*, were expressed in distinct continuous domains and displayed a typical "saturating" phenotype (Supplementary Fig. 2h). In order to find out whether any of these transcription factors were capable of repressing window genes, we individually knocked them down (Supplementary Fig. 3a–c), incubated the knockdown embryos either in AZK or in DMSO and compared the expression of two well-characterized "window" genes *Wnt1* and *Wnt2*[4] in the knockdowns at late gastrula stage. Knockdowns of *Svb*, *Dac* and *NVE11868* led to no significant change in the expression of *Wnt1* and *Wnt2* in comparison to control shRNA (Supplementary Fig. 3d). Therefore, these genes were also excluded from further analyses, and we focused on the remaining four candidates, *Bra*, *FoxA*, *FoxB*, and *Lmx*, and characterized their mutual expression domains and the effect of their knockdowns upon normal and pharmacologically enhanced β-catenin signaling (See also Supplementary Results and Discussion 1).

dpf *APC⁻/⁻* embryos (Fig. 1e; Supplementary Fig. 2a–f), and controls. We then identified all putative transcription factor-coding genes upregulated by elevated β-catenin in all treatments by comparing our lists of differentially expressed genes with the list of gene models with a predicted DNA binding domain. We found twelve such putative transcription factors (Fig. 1f,

**Repressor X is not a single gene but a unit of four genes.** The area of strong *Bra* expression overlaps with the *Wnt1* expression domain and abuts the *Wnt2* expression domain (Fig. 2). Upon

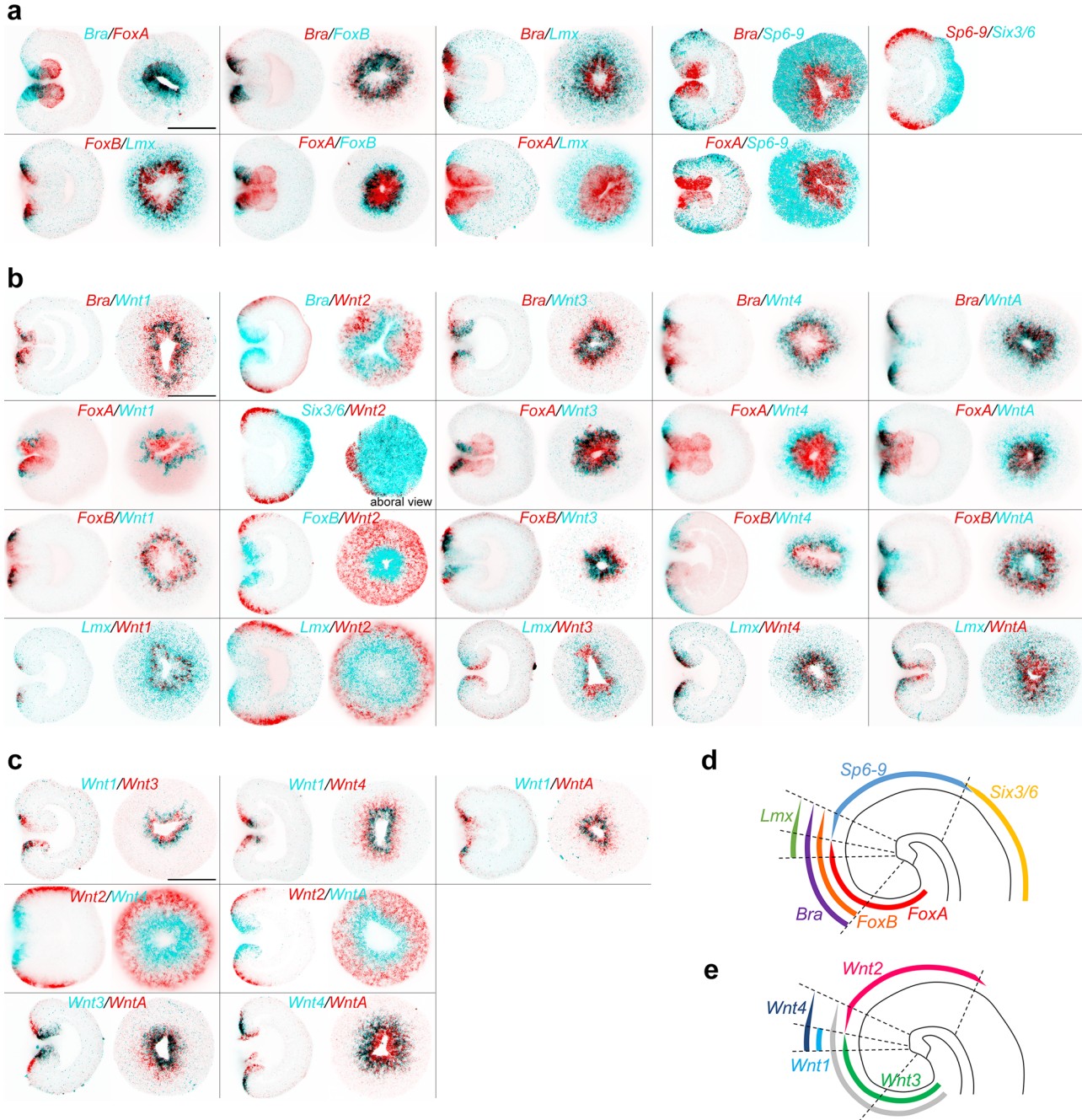

**Fig. 2 Double FISH analysis of the expression domains of the four main repressor X candidates, oral *Wnt* genes, midbody markers *Wnt2* and *Sp6-9*, and aboral marker *Six3/6*. a** FISH analysis of the expression domains of the transcription factor genes *Bra*, *FoxA*, *FoxB*, *Lmx*, *Sp6-9* and *Six3/6* in relation to each other. **b** FISH analysis of the expression domains of the abovementioned transcription factor genes in relation to the expression domains of the ectodermally expressed *Wnt* genes. **c** FISH analysis of the expression domains of the ectodermally expressed *Wnt* genes in relation to each other. **d** Schematic representation of the expression boundaries of the transcription factors in the *Nematostella* gastrula. **e** Schematic representation of the expression boundaries of the *Wnt* genes in the *Nematostella* gastrula. On **a–c**, lateral views (oral to the left) and oral views (unless specified otherwise) of representative embryos from two independent experiments with n > 30 for each combination of in situ hybridization probes are shown. Scale bars 100 μm. Dashed lines on **d** and **e** represent the same molecular boundaries.

*Bra* knockdown, *Wnt1* expression was abolished not only in the AZK treatment but also in the DMSO treated controls, suggesting that *Wnt1* is positively regulated by *Bra* (Fig. 3, Supplementary Fig. 4). In contrast, *Wnt2* expression domain expanded orally in the DMSO controls and became ubiquitous upon the AZK treatment (Fig. 3, Supplementary Fig. 4). This suggests that Brachyury acts as the hypothetical transcriptional repressor X for *Wnt2*, but not for *Wnt1*. *FoxA* is expressed in the future pharynx

of the embryo and in the domain immediately around the blastopore inside the ring of *Wnt1* expressing cells (Fig. 2). *FoxA* knockdown did not affect *Wnt2* expression, but *Wnt1* expression became stronger and expanded further orally in DMSO and globally in the AZK treatment (Fig. 3, Supplementary Fig. 4a). Thus, FoxA appears to act as the hypothetical repressor X for *Wnt1*, but not for *Wnt2*. *FoxB* is co-expressed with *Bra* in the domain where *Bra* expression is strong, i.e. abutting the *Wnt2*

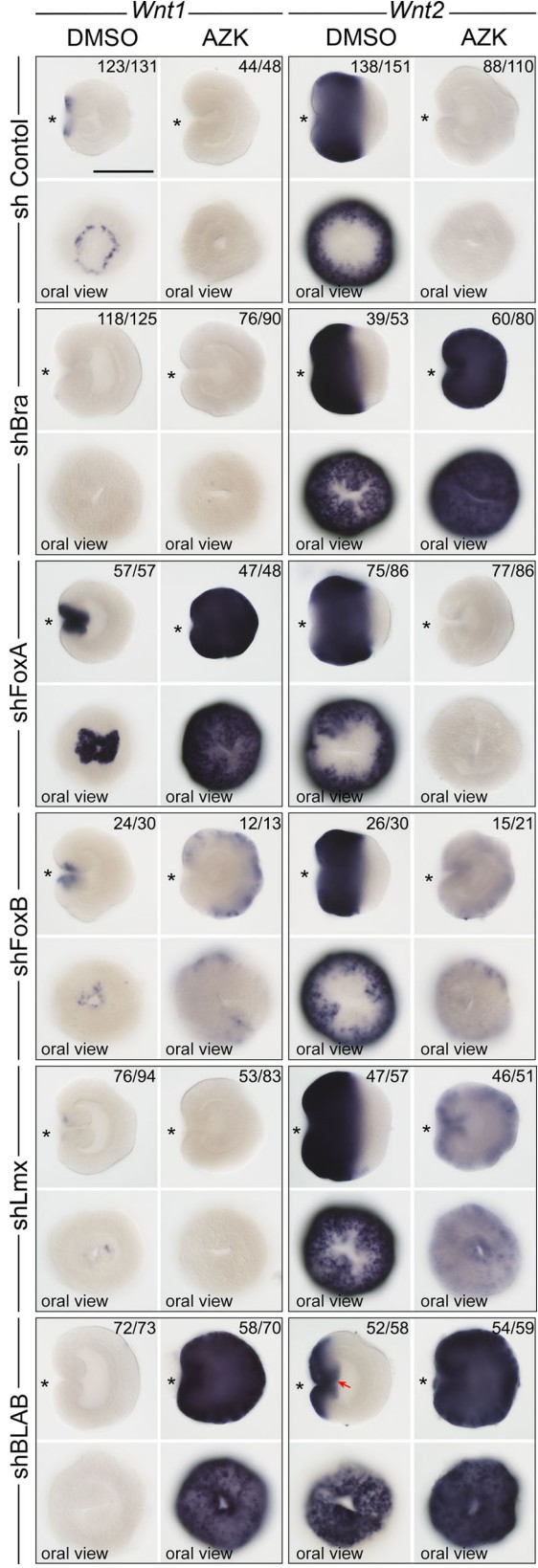

**Fig. 3 The effect of the repressor X candidates knockdown on the expression of the "window" genes *Wnt1* and *Wnt2*.** *Bra* and *Lmx* knockdowns convert *Wnt2* into a "saturating" gene, while *FoxA* knockdown does the same with *Wnt1*. The effect of *Lmx* knockdown appears to be similar but weaker than that of *Bra*. *FoxB* knockdown results in a "weak AZK effect" on both *Wnt1* and *Wnt2* suggesting that FoxB mildly represses both. The effects of the knockdowns of *Bra*, *Lmx*, and *FoxB* on *Wnt2* expression are non-redundant, but similar and additive (see Supplementary Figs. 6 and 7). Quadruple knockdown with shRNA against *Bra*, *Lmx*, *FoxA* and *FoxB* (=shBLAB) removes oral molecular identity of the embryo completely. Red arrow indicates the bottom of the pharynx expressing the midbody marker *Wnt2*. On lateral views, asterisk denotes the blastopore. The numbers in the top right corner show the ratio of embryos displaying the phenotype shown on the image to the total number of embryos treated and stained as indicated on the figure. Scale bar 100 μm.

(Fig. 3, Supplementary Fig. 4a). Single, double and triple knockdown experiments suggest that the role of these two transcription factors appears to be in supporting the activity of Bra and FoxA in the areas, where they are co-expressed (Supplementary Results and Discussion 1, Supplementary Figs. 5–7). Simultaneous knockdown of *Bra*, *Lmx*, *FoxA* and *FoxB* with a mixture of shRNAs (shBLAB) completely abolishes the oral identity of the embryo at the molecular level: the midbody marker *Wnt2* shifts orally, expanding all the way to the bottom of the pharynx in DMSO, while the *Wnt2*-free aboral domain expands (Fig. 3, Supplementary Fig. 5). A much more pronounced expansion of the aboral domain and the confinement of the midbody marker *Wnt2* to the oralmost part of the embryo upon the combined knockdown of *Bra* together with either *Lmx* or *FoxB* or both in comparison to the individual *Bra* knockdown shows that the functions of these genes are non-redundant (Fig. 3; Supplementary Results and Discussion 1, Supplementary Figs. 5–7). We conclude that oral "window" genes are activated by β-catenin signaling (either directly or indirectly), and repressed by β-catenin-dependent "saturating" transcription factors. No single transcriptional repressor X exists, but rather *Bra*, *Lmx*, *FoxA* and *FoxB* appear to be the unit defining oral identity in the *Nematostella* embryo. Strikingly, the knockdown of any of these four transcription factors did not prevent normal gastrulation, and all the effects at this developmental stage remained purely molecular, pointing at the potential role of maternal factors in the gastrulation process (see Suppl. Results and Discussion 2–3, Supplementary Figs. 8, 9).

**Repressor X regulatory logic applies to the whole O–A axis.** Previous work demonstrated that the aboral markers *FoxQ2a* and *Six3/6*, which are downregulated upon elevated β-catenin signaling, still require some β-catenin signaling in order to be expressed[9], i.e. they may also be window genes. Therefore, it is conceivable that the patterning logic we discovered for the oral domain may be applicable to the whole of the O–A patterning, with more orally expressed β-catenin-dependent genes acting as transcriptional repressors for the more aborally expressed β-catenin-dependent genes. To test that, we investigated the mechanism of the maintenance of the other clear molecular boundary present in late gastrula ectoderm: the one between the *Wnt2*-positive midbody domain and the *Six3/6*-positive aboral domain (Fig. 1b). If the proposed regulatory logic were correct, there would have to exist at least one "transcriptional repressor Y", which: (i) has to be expressed in the midbody domain, (ii) has to counteract the oral expansion of the aboral domain, and (iii) has to be positively regulated by β-catenin and repressed by the oral, "saturating" transcription factors (i.e. it has to be encoded by a "window" gene). Since "window" genes are downregulated upon

expression domain, and *Lmx* is a weakly expressed gene active in a domain starting from the *Wnt1* expressing cells and quickly fading out further aborally (Fig. 2). *FoxB* knockdown resulted in the expansion of both *Wnt1* and *Wnt2* expression in AZK, but the staining appeared weak, and *Lmx* RNAi effect on *Wnt1* and *Wnt2* largely recapitulated the effect of *Bra* RNAi, albeit milder

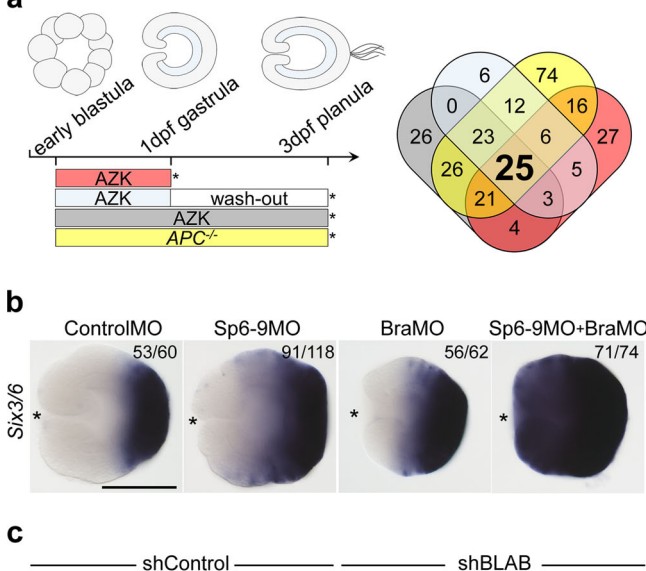

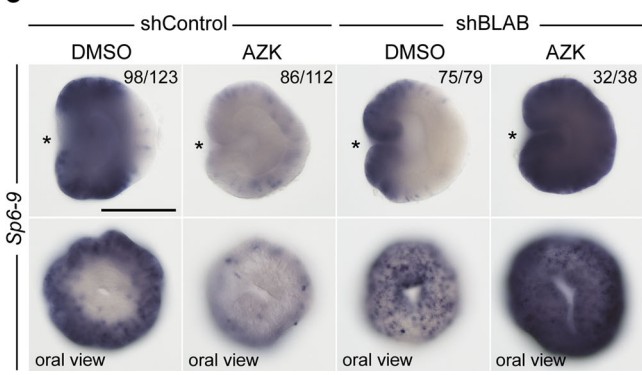

**Fig. 4 Midbody domain prevents oral expansion of the aboral domain.**
**a** Scheme of the treatments and Venn diagram showing the number
of putative transcription factors downregulated by various treatments.
**b** Sp6-9 prevents oral expansion of the aboral marker *Six3/6*. In BraMO,
*Six3/6* expression is also expanded orally, likely due to the oral shift of the
*Sp6-9* expression upon *Bra* knockdown (see Supplementary Fig. 5). Oral
expansion of *Six3/6* is enhanced upon double knockdown of *Sp6-9* and
*Bra*. Lateral views, oral to the left; asterisk denotes the blastopore. **c** *Sp6-9* is
a "window" gene shifting orally upon simultaneous knockdown of *Bra, Lmx,
FoxA* and *FoxB* (=shBLAB) and expanding globally upon shBLAB knockdown
followed by AZK treatment. *Sp6-9*-free area disappears in shBLAB. Lateral
views, oral to the left; asterisk denotes the blastopore. The numbers in the
top right corner on **b, c** show the ratio of embryos displaying the phenotype
shown on the image to the total number of embryos treated and stained as
indicated on the figure. Scale bars 100 μm.

elevated β-catenin signaling, we looked at the transcription factor
coding genes downregulated by all treatments in our RNA-Seq
experiment, and found 25 such genes (Fig. 4a). We performed
in situ hybridization with probes against all of them and excluded
18 genes expressed either in single ectodermal cells, endoder-
mally, or whose expression domain included the aboral pole
(Supplementary Fig. 10, Supplementary Table 2). Thus, we were
left with seven transcriptional repressor Y candidates expressed in
the midbody but not in the aboral domain: *Sp6-9, Nk1, Dlx,
MsxA, FoxG1, Rx,* and *HES-like* (Supplementary Fig. 10, Sup-
plementary Table 2). In order to test whether they were capable of
counteracting the oral expansion of the aboral domain, we per-
formed individual knockdowns of all of them followed by in situ
hybridization against the aboral marker *Six3/6* (Supplementary
Fig. 11). Out of all candidates, only the knockdown of the gene
encoding the Krüppel-like transcription factor Sp6-9 resulted in

the oral expansion of the *Six3/6* expression domain. (Fig. 4b,
Supplementary Fig. 12a, b, Suppl. Results and Discussion 1 and
3). Predictably, since *Bra* knockdown results in the oral shift of
the midbody domain (Fig. 3, Supplementary Fig. 5) and expan-
sion of the aboral domain (Fig. 4b), *Six3/6* expansion was much
more pronounced upon the combined knockdown of *Sp6-9* and
*Bra* (Fig. 4b). Finally, we tested whether *Sp6-9* fulfilled the
remaining transcriptional repressor Y criterion set above, namely
whether it was a "window" gene. We could show that the
knockdown of the four oral transcription factors *Bra, FoxA, FoxB*
and *Lmx* expanded the expression of *Sp6-9* orally in DMSO and
globally in AZK (Fig. 4c), i.e. *Sp6-9* behaved as a "window" gene.
Curiously, in addition to the broad expression in the midbody
domain (bordering the *Bra* domain orally and the *Six3/6* domain
aborally; Fig. 2), *Sp6-9* is also strongly expressed in individual
cells scattered all over the embryo. This single-cell expression was
not affected by the modulation of the β-catenin signaling
(Fig. 4c). Taken together, Sp6-9 appears to act as hypothetical
repressor Y at least for *Six3/6*, which suggests that the regulatory
logic we proposed is applicable not just to the oral domain but to
the whole β-catenin-dependent O–A axis patterning in the
*Nematostella* ectoderm.

**Aboral identity represents the default state.** We demonstrated
that the logic of the β-catenin-dependent O–A patterning relied
on more orally expressed β-catenin targets displacing the
expression domains of the more aborally expressed β-catenin
targets further aborally. Therefore, we decided to test whether
aboral fate represented the default state of the whole *Nematostella*
embryo, which then became progressively restricted to the aboral
domain by the orally expressed β-catenin-dependent factors, as
it is described for the anterior ectodermal domain in
deuterostomes[13–15,26]. The fact that the major aboral determinant
*Six3/6* requires an initial β-catenin signal in order to be
expressed[9] may be used as evidence against this hypothesis.
However, *Six3/6* is a zygotic gene, whose expression becomes
detectable at 12 h post fertilization (hpf), which is 4 h later than
the onset of expression of the oral marker *Bra* (Fig. 5a). Notably,
even the earliest expression of *Six3/6* is not ubiquitous, but
localized to the future aboral side of the O–A axis. However, we
do find aboral markers, whose expression is initially maternal and
ubiquitous and subsequently becomes restricted to the aboral end
in a β-catenin-dependent manner. One of them is *Frizzled 5/8*
(Fig. 5a, b), which was shown to be a negative regulator of *Six3/6*
and *FoxQ2a* in *Nematostella* and sea urchin[9,15,27]. The other one
is *SoxB1* (Fig. 5a, b), whose initially ubiquitous expression is
cleared β-catenin-dependently out of the organizer and endo-
mesodermal area in deuterostomes[28,29]. Individual or simulta-
neous knockdowns of the oral and midbody factors *Bra* and *Sp6-9*
in *Nematostella* significantly expand the expression domain of
*SoxB1* (Fig. 5c). Although qPCR data suggest that sea urchin
*SoxB1* is a positive maternal upstream regulator of *FoxQ2*[30], the
negative effect of *SoxB1* knockdown on *Six3/6* and *FoxQ2a*
expression in *Nematostella* is not pronounced (Supplementary
Fig. 13), and it is still unclear what kind of positive regulatory
input maintains the aboral expression of *Six3/6* and hence other
aboral markers. Nevertheless, our data clearly support the aboral-
by-default model.

**Endoderm is not a prerequisite for the ectodermal patterning.**
In many investigated bilaterians, the earliest function of β-catenin
signaling is to define the endomesodermal territory, and its role in
the P–A patterning appears to kick in later[13–15,23]. We were
interested to see whether this was also the case in *Nematostella*.
Previous work showed that *Nematostella* embryos failed to form

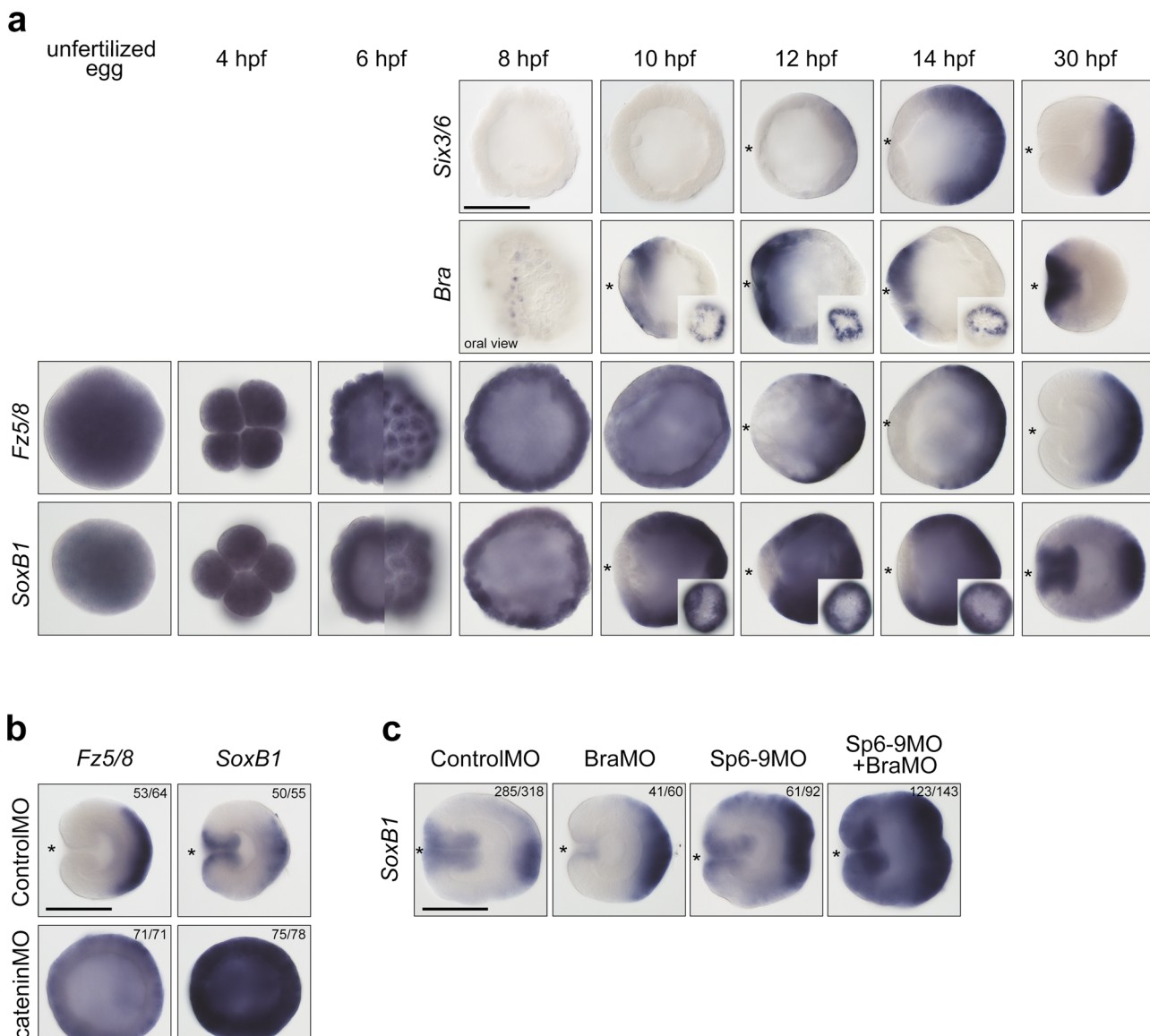

**Fig. 5 *Nematostella* embryo initially has aboral identity, which later becomes restricted to the aboral domain. a** *Six3/6* is detectable in the aboral portion of the embryo from 12 h post fertilization (hpf) on. *Bra* becomes detectable in a group of cells on the future oral side of the embryo as early as 8 hpf, and by 10 hpf it forms a ring around the future preendodermal plate. *Fz5/8* is a maternally deposited transcript. *Fz5/8* expression shifts to the future aboral side by 12 hpf. *SoxB1* is also a maternally deposited transcript. The loss of *SoxB1* staining in the future endodermal territory occurs simultaneously with the formation of the *Bra* ring, and is likely regulated by the same mechanism. By gastrula stage, *SoxB1* is expressed in the blastopore lip and aborally. On all lateral views, on which the O–A axis is discernible, the oral end is marked with an asterisk. Inset images of 10, 12 and 14 hpf embryos stained for *Bra* and *SoxB1* show the lack of expression in the putative preendodermal plate on embryos orientated with their oral ends facing the viewer. 6 hpf images of *Fz5/8* and *SoxB1* expression show the optical midsection (left) and the surface view (right) of the same embryos. **b** *Fz5/8* and *SoxB1* expression remains ubiquitous in the β-catenin morphants. Lateral views of the 30 hpf gastrulae, oral ends are marked with an asterisk. **c** *SoxB1* expression upon *Bra* knockdown appears weaker in the oral domain and expanded in the aboral domain, which is likely due to the oral shift of the *Sp6-9* expression. *Sp6-9* knockdown significantly expands *SoxB1* expression fusing the oral and aboral expression domains. Simultaneous knockdown of *Bra* and *Sp6-9* makes this effect even more pronounced consistent with the general aboralization of the embryo. The numbers in the top right corner on **b**, **c** show the ratio of embryos displaying the phenotype shown on the image to the total number of embryos treated and stained as indicated on the figure. Scale bars 100 μm.

preendodermal plates and gastrulate when β-catenin signaling was suppressed by *cadherin* mRNA overexpression[8] or β-catenin morpholino injection[9]. The lack of gastrulation clearly suggested that the role of β-catenin signaling in the determination of the endomesoderm was conserved since before the cnidarian–bilaterian split[8,31]. In β-catenin morphants, not only the formation of the preendodermal plate, but also the expression of the "saturating" genes responsible for patterning the oral ectoderm such as *Bra*, *FoxA* and *FoxB* is abolished[9]. Strikingly, the embryos placed in 5 μM AZK shortly after fertilization (2 hpf) also fail to form preendodermal plates and remain spherical. However, these embryos, unlike β-catenin morphants, express *FoxA* and *FoxB* ubiquitously[9]. In contrast, in our 5 μM AZK incubation experiments starting at 10 hpf, gastrulation process was not affected, and "saturating" oral ectodermal markers were ubiquitously expressed in the ectoderm but never extended into

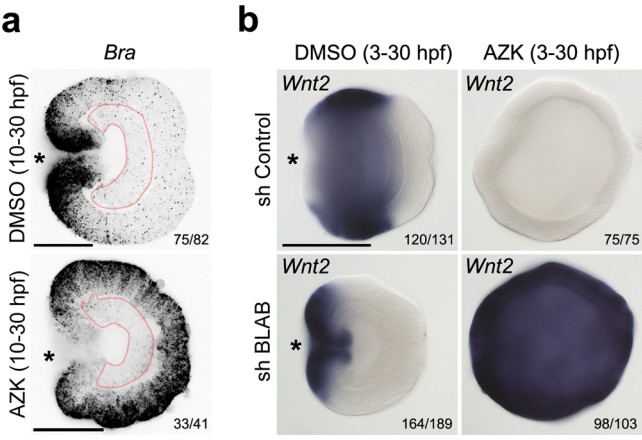

**Fig. 6 Endoderm has no influence on O–A patterning of the ectoderm.**
**a** Fluorescence in situ hybridization shows that *Bra* expression does not extend into the endoderm of the embryos (pink outline), which were placed in 5 μM AZK after the time of the specification of the endodermal domain. **b** 5 μM AZK incubation starting before the time of the specification of the endodermal domain prevents endoderm formation but still leads to the abolishment of *Wnt2* expression in shControl and to the conversion of *Wnt2* into a "saturating" gene upon shBLAB knockdown (compare with Fig. 3). The numbers in the bottom right corner show the ratio of embryos displaying the phenotype shown on the image to the total number of embryos treated and stained as indicated on the figure. Scale bars 100 μm.

the endoderm (Fig. 6a) suggesting that the definition of the endodermal territory was complete prior to the onset of the treatment. Since the "saturating" expression behavior of the oral ectoderm markers was observed independent of the presence or absence of the endoderm, we asked whether the same was true for a "window" gene *Wnt2*. We showed that AZK treatment of the shBLAB-injected and control embryos starting at 3 hpf suppressed endoderm formation and that simultaneous knockdown of *Bra*, *FoxA*, *FoxB* and *Lmx* (shBLAB) followed by AZK treatment resulted in ubiquitous expression of *Wnt2*. Thus, the knockdown of the four "saturating" genes controlling the development of the oral domain resulted in the "saturating" expression of the "window" gene *Wnt2* both in the absence (Fig. 6b) and in the presence (Fig. 3) of the endoderm. This suggests that, similar to Bilateria, the roles of β-catenin signaling in defining the endodermal territory and ectodermal patterning in *Nematostella* are separable in time, and that the presence or absence of the endoderm does not influence ectodermal patterning at least until 30 hpf when the embryos were fixed and assayed.

## Discussion

As a bilaterian sister group, cnidarians provide us with a key reference point regarding the evolution of body axes patterning and germ layer formation. Like in ambulacrarian deuterostomes, the definition of the future endoderm in *Nematostella* appears to be the earliest patterning event and relies on β-catenin signaling. Since both, morpholino knockdown of β-catenin and AZK-mediated stabilization of β-catenin at 2–3 hpf lead to the failure of the preendodermal plate formation[9], it appears plausible that a certain precise dose of β-catenin signaling is required for the specification of the endodermal territory. Successful gastrulation of the embryos treated with AZK after 10 hpf suggests that the prospective endoderm is already specified by this time and that, once defined, the endoderm becomes insensitive to β-catenin signaling modulation at least until late gastrula stage. The expression of the genes patterning *Nematostella* ectoderm begins

after the specification of the endodermal territory, and their "window" or "saturating" behavior in response to AZK is not dependent on the presence or absence of the endoderm. In several investigated bilaterians, the early β-catenin signal defining the endomesoderm appears to rely on maternal components[21,32–34]. In the future, it will be important to test how the switch from the β-catenin signaling-dependent specification of the endodermal domain to the β-catenin signaling-dependent ectodermal patterning in *Nematostella* relates to the activation of the zygotic transcription, which has been reported to occur at some point between 2 and 7 hpf[35]. Curiously, the canonical β-catenin-dependent deuterostome endomesodermal markers *Bra* and *FoxA*[13,14,22,36–41] are never expressed in the preendodermal plate of *Nematostella*. Instead, they are markers of the blastopore lip, i.e., of the oral ectoderm, which gives rise to the pharynx of the animal. In contrast, the expression signature and the response of the preendodermal plate to β-catenin signaling is reminiscent of the mesodermal domain in the echinoderm embryos[9,22,32,42]. This provides some additional support to the hypothesis that the anthozoan endoderm and pharyngeal ectoderm may be homologous to the bilaterian mesoderm and endoderm, respectively[43].

Our data also allow re-evaluating the possible correspondence of the cnidarian and bilaterian body axes. In addition to the main, O–A body axis patterned by β-catenin signaling, anthozoans have a second, so-called "directive" axis patterned by BMP signaling[44–46], which is strikingly similar to the situation in Bilateria, where the P–A axis is patterned by Wnt/β-catenin signaling, and the dorsal-ventral axis is patterned by BMP signaling. The similarity can have two possible explanations: either the last common ancestor of Cnidaria and Bilateria was bilaterally symmetric, in which case bilaterality must have been lost in radially symmetric medusozoan cnidarians, or anthozoan Cnidaria and Bilateria evolved bilaterally symmetric body plans independently but used the same signaling pathways for symmetry breaking and patterning[2]. If bilaterality indeed evolved prior to the cnidarian–bilaterian split, the direct correspondence of the anthozoan and bilaterian body axes can be explained by three alternative, extensively debated scenarios. In the first scenario (O–A = A–P, Fig. 7a), the O–A axis is proposed to correspond to the anterior-posterior axis of Bilateria. The proponents of this scenario stress the importance of the direct correspondence of the animal-vegetal axis of the egg to the O–A axis in cnidarians, and the conservation of the origin of the mouth from the animal hemisphere material in Cnidaria, most Protostomia and Deuterostomia. They argue that once the gastrulation site switched from the animal to the vegetal pole at the base of Bilateria, the change in the position of the blastopore did not affect the location of the mouth and other structures. Therefore, it was suggested that cnidarian and bilaterian apical plates—the neurogenic territories developing at the vegetal pole in Cnidaria and at the animal pole in Bilateria—are non-homologous[31]. Finally, the role of the "anterior" *Hox* gene *Anthox6/HoxA* in the development of the oral end and the "non-anterior" *Hox* gene *Anthox1/HoxF* in the development of the aboral end of the *Nematostella* embryo[45,47] has been seen as a supporting argument for the O–A = A–P scenario. However, *HoxA* and *HoxF* are expressed in non-adjacent domains in the embryo in different germ layers[47], and are located on different chromosomes in the genome[48], in contrast to the genomically linked *Hox* genes, which are expressed in staggered domains and generate a Hox code patterning the second, directive axis under BMP control[46,48,49]. Another piece of evidence against the O–A = A–P scenario is that the apical ectodermal domains opposing the gastrulation sites both in Cnidaria and Bilateria have a strikingly similar expression signature making the homology of the cnidarian and bilaterian

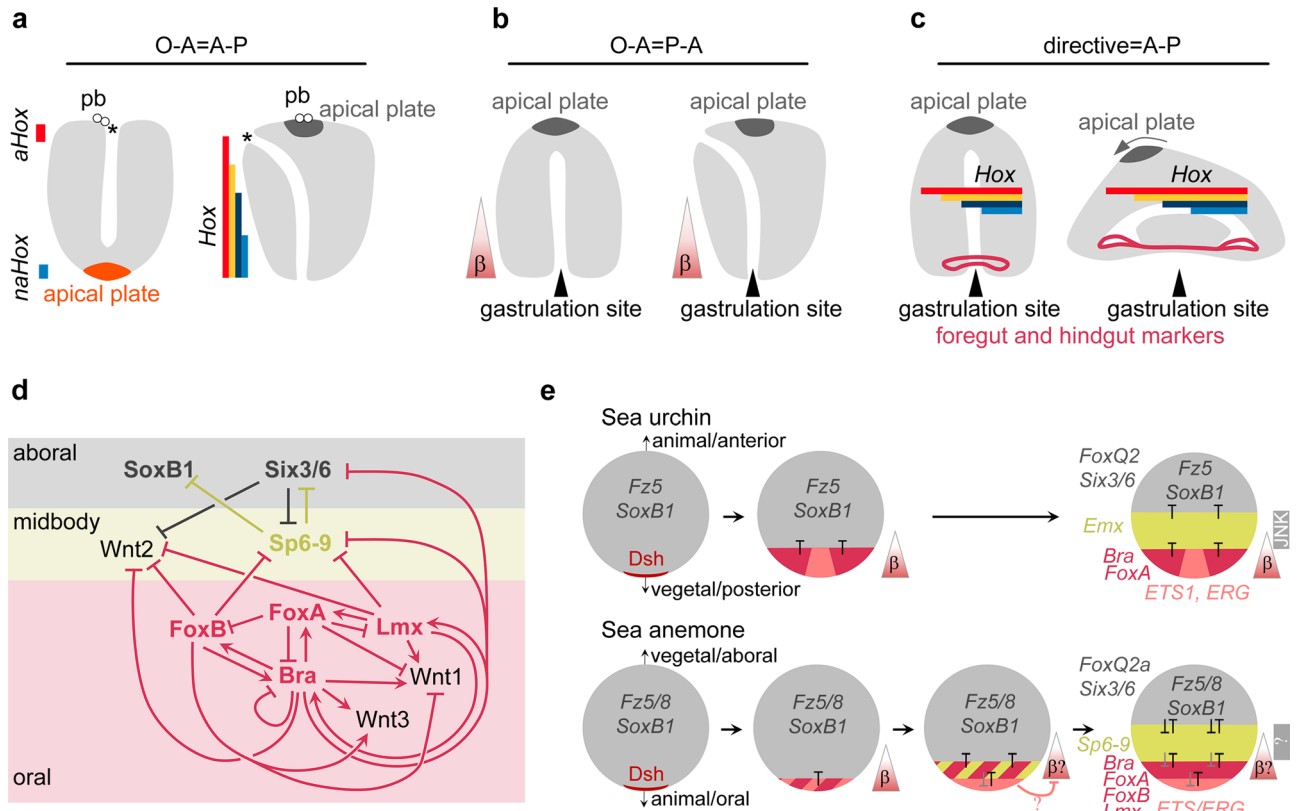

**Fig. 7 Oral–aboral patterning regulation in *Nematostella* and P-A patterning in sea urchin are comparable. a–c** Scenarios of the direct correspondence of the cnidarian and bilaterian body axes. pb – polar bodies, *aHox* – anterior *Hox* gene, *naHox* – non-anterior *Hox* gene, asterisk denotes the mouth. Triangles with a β denote the direction of the β-catenin signaling gradient. **d** Putative topology of the gene regulatory network of the β-catenin-dependent O–A patterning in *Nematostella*. The GRN explains why the midbody domain does not expand into the oral and into the aboral domains, and why the aboral domain does not expand into the midbody. It does not explain, however, why the oral domain does not expand aborally. **e** Comparison of the early β-catenin-dependent patterning in sea urchin and *Nematostella* shows clear similarities. Unfertilized egg with maternal *Fz5/8* and *SoxB1* mRNA (future anterior/aboral markers) and maternal Dsh protein localized at the gastrulation pole[65,66]. Upon activation of β-catenin signaling in the embryo, first in the endomesodermal domain and then in the posterior/oral ectoderm the expression of *Fz5/8* and *SoxB1* is suppressed, and the anterior/aboral markers (including the zygotic genes *Six3/6* and *FoxQ2*) become progressively confined to one side of the axis. The axis becomes patterned by mutually repressive transcription factors (T). Gray "T" in *Nematostella* indicate repressive interactions, for which candidate transcription factors are not known. Triangles with a β denote the direction of the β-catenin signaling gradient. β? indicates that in *Nematostella*, nuclear β-catenin could only be experimentally detected until midblastula stage[9], after which the presence of nuclear β-catenin gradient is deduced based on target gene response. After preendodermal plate is specified in *Nematostella*, β-catenin signaling becomes repressed there by an unknown mechanism[9], possibly involving ERG[42].

apical plates highly plausible[50–52]. Also, the oral end of cnidarians is characterized by a β-catenin signaling maximum, which also appears to be a conserved feature of the posterior rather than the anterior end both in protostome and in deuterostome Bilateria. Thus, the second scenario (O–A = P–A, Fig. 7b) suggests that the O–A axis of Cnidaria corresponds to the posterior–anterior axis of Bilateria. The O–A = P–A scenario, however, does not consider the importance of the *Hox*-dependent axial patterning in Anthozoa. The third scenario (directive = A–P, Fig. 7c) proposes that the directive axis of anthozoans may correspond to the anterior-posterior axis of the ancestral bilaterian, whose blastopore closed in an amphistomic, slit-like fashion generating a mouth and an anus at opposing ends connected by a through gut. This scenario is supported by the circumblastoporal expression of several bilaterian foregut and hindgut markers in Cnidaria and by the role of the staggered expression of *Hox* genes in patterning the directive axis in *Nematostella* and the A–P axis in Bilateria[49,51,52]. The directive=A–P hypothesis is somewhat hampered by the unclear orthology of the cnidarian and bilaterian *Hox* genes, their likely independent diversification in Cnidaria and Bilateria, and their expression along the body axis patterned by BMP signaling

and under BMP control in *Nematostella*[2,46], which is highly unusual for Bilateria.

Although none of the three scenarios above explains the correspondence of the two anthozoan and two bilaterian body axes without contradiction, we can assess whether any of them is supported by our new data on the mechanism of the β-catenin-dependent patterning of the main cnidarian body axis better than the others. Here we showed that *Bra*, *FoxA*, *Lmx* and *FoxB* define the oral molecular identity of the *Nematostella* embryo and prevent oral expansion of the more aborally expressed β-catenin targets (Fig. 7d). We also identified *Sp6-9*, a "window" gene expressed in the midbody domain, as the agent preventing the oral expansion of the aboral domain (Fig. 7d). The whole *Nematostella* embryo initially represents an aboral ectodermal territory, which is established maternally (Fig. 7e). During the first day of development, this territory becomes restricted to the aboral end of the O–A axis in a β-catenin-dependent manner by "saturating" and "window" transcriptional repressors, which form mutually repressive pairs capable of generating sharp domain boundaries (Fig. 7e). This is highly similar to the situation demonstrated in non-chordate deuterostomes like echinoderms and hemichordates[13–15,28,32].

Comparison with sea urchin reveals remarkable conservation of the components of the axial patterning gene regulatory network downstream of β-catenin. In sea urchin, *Bra* and *FoxA* are central in the β-catenin-dependent axial patterning of the blastoporal domain[40,53]. The midbody domain of the sea urchin embryo appears to be defined by an Antennapedia class transcription factor *Emx*[54,55], rather than by a Krüppel-like factor *Sp6-9*. However, the patterning of the apical ectoderm is again accomplished by the same components in the sea anemone and sea urchin embryos[15,50]. Importantly, not only the genes involved, but also the regulatory logic of gradual restriction of the apical ectodermal territory by β-catenin-dependent transcription factors appears to be highly similar in the β-catenin-dependent O–A patterning of the ectoderm in the anthozoan *Nematostella* and in the posterior–anterior patterning in sea urchin and other investigated deuterostomes, including vertebrates[11,13–16,40,53] (Fig. 7e), while the situation in protostomes appears to be more derived. Based on this remarkable similarity we conclude that the processes of ectodermal patterning of the cnidarian O–A axis and the deuterostome P–A patterning share a common evolutionary origin predating the cnidarian–bilaterian split. Thus, independent of whether the second, BMP-dependent body axes of anthozoans and bilaterians evolved independently or not, we propose that the cnidarian O–A and the deuterostome P–A body axes are likely homologous (O–A = P–A).

## Methods

**Animals, microinjection, APC mutants, and transplantations**. *Nematostella* polyps were kept in *Nematostella* medium (16‰ artificial seawater, NM) at 18 °C in the dark and induced to spawn by placing them into a 25 °C, illuminated incubator for 10 h. The eggs were fertilized for 30 min and dejellied in 3% L-cysteine/NM and washed 6 times in NM. Microinjection was performed under the Nikon TS100F microscope using Eppendorf Femtojet and Narishige micromanipulators. The *APC* mutant line was generated by injecting *Nematostella* zygotes with 500 ng/µl single gRNAs (protospacer 5′CACAGCTATGAGGGCCAC) and 500 ng/µl nls-Cas9 (PNA Bio, Thousand Oaks, CA, USA). Mosaic F0 animals were crossed to produce *APC*[+/-] F1 carrying a single T insertion after the position 331 of the coding sequence of *Nematostella APC* (Genbank KT381584). Heterozygous F1 were crossed to obtain F2. In situ hybridization analysis showed that 27% of the F2 embryos expressed *Bra* throughout the ectoderm of the gastrula, while 73% had normal *Bra* expression ($N$ = 221). At 3 dpf, 10 out of 10 randomly selected F2 embryos demonstrating the typical bagel phenotype similar to that of the AZK treated embryos proved to be *APC*[−/−] when genotyped by Sanger sequencing of the mutated locus (Supplementary Fig. 1d). For genotyping live polyps, individual primary polyps or tentacle clips were fixed by 3 washes in 100% methanol, aspirated, and dried for 20 min at 50 °C with the tube lids open. Then, samples were digested in 30 µl of extraction buffer (10 mM Tris-HCl pH8, 1 mM EDTA pH8, 25 mM NaCl, 200 µg/ml proteinase K) for 2 h at 50 °C, and proteinase K was inactivated by heating the samples to 95 °C for 5 min. After proteinase K inactivation, 3 µl of the digest was used as template for the PCR with the primers flanking the locus recognized by the gRNA (APCspF 5′AGAATCCTGCA GAAGATGAACA, APCspR 5′CCTGGCATACAAAGGTGACA). The PCR product was purified and directly sequenced with the APCspF primer. For genotyping embryos after in situ hybridization, the embryos were dehydrated in ethanol series, washed twice with 100% ethanol, embedded into Murray's clear solution (benzyl benzoate:benzyl alcohol = 2:1), imaged, washed several times in 100% methanol and then processed as described above. During experiments, the embryos were kept in the 21 °C incubator. Blastopore lip transplantations were performed as described[4]. The significance of the difference in the transplantation outcomes was assessed by performing the Z score test for two population proportions (https://www.socscistatistics.com/tests/ztest/default.aspx).

**Pharmacological treatments, gene knockdown, overexpression**. 1-azakenpaullone (Sigma) used for the treatments was prepared by diluting 5 mM AZK dissolved in DMSO with NM. Equal volume of DMSO was used to treat the control embryos. 5 µM AZK was used for treating the embryos used for the RNA-Seq experiments as well as for the transcriptional repressor X and Y search. The time windows of the treatments are presented in Fig. 1e; briefly, unless specified otherwise, the embryos were incubated in AZK or DMSO from 10 hpf (early blastula) until either 30 hpf (late gastrula) or 72 hpf (3 dpf planula larva). For the embryos incubated from 10 until 30 hpf, RNA was extracted either immediately at 30 hpf or after a wash-out and a 42 h long incubation in NM (i.e., at 72 hpf). Gene knockdowns were performed by electroporation with shRNA as specified[49,56]. Two non-overlapping shRNAs were used for each of the genes to confirm the specificity of the observed phenotypes except for the cases of *Brachyury*, *Sp6-9*, *Nk1*, and *Dlx*, where two or one shRNAs and one translation-blocking morpholino (MO) were used (Supplementary

Tables 3–4). shRNA against *mOrange* was used as a control for all other shRNA, and a control MO we described previously[4] (Supplementary Table 4) was used to control for the BraMO, Nk1MO, DlxMO, and Sp6-9MO phenotypes. The RNAi efficiency was estimated by in situ hybridization and Q-PCR (Supplementary Fig. 3a, b, Supplementary Fig. 11a, b, Supplementary Table 5), and the activity of the morpholinos was assayed by co-injecting them with the wild type and 5-mismatch mRNA containing the morpholino recognition sequences fused to mCherry (Supplementary Fig. 3c, Supplementary Fig. 11c). Capped mRNA was synthesized using mMessage mMachine kit (Life Technologies) and purified with the Monarch RNA clean-up kit (NEB). *Bra* and *FoxB* mRNA for overexpression was also produced as described above. A stabilized form of β-catenin was generated by removing the first 240 bp of the β-catenin coding sequence as described in[57]. An ATG was added, and the fragment, which we called β-cat_stab, was cloned into an expression vector downstream of the ubiquitously active *EF1α* promoter[43]. Mosaic expression of the *EF1α::β-cat_stab* was achieved by meganuclease-assisted transgenesis, as described[58]. Primers against GAPDH were used as normalization control in QPCR.

**Transcriptome sequencing and analysis**. Total RNA was extracted with TRIZOL (Life Technologies) or with GeneElute Mammalian Total RNA Miniprep Kit (Sigma) according to the manufacturer's protocol; poly-A enriched mRNA library preparation (Lexogen), quality control, and multiplexed Illumina HiSeq2500 sequencing (50 bp, single-end) were performed at the Vienna BioCenter Core Facilities. The number of the sequenced biological replicates of different treatments is shown in the Supplementary Fig. 2. SAMtools 1.11[59] was used for format conversion. The reads were aligned with STAR[60] to the *Nematostella vectensis* genome[61] using the ENCODE standard options, with the exception that–alignIntronMax was set to 100 kb. Hits to the gene models v2.0 (https://figshare.com/articles/Nematostella_vectensis_transcriptome_and_gene_models_v2_0/807696) were tallied with featureCounts[62], and differential expression analysis was performed with DeSeq2[63]. Expression changes in genes with Benjamini-Hochberg adjusted p-value < 0.05 were considered significant. No additional expression fold change cutoff was imposed. Transcription factor candidates were determined by analyzing the transcriptome with INTERPROSCAN[25] and filtering for genes containing the domains described in[64]. The intersection between the latter set and our differentially expressed genes comprised the models of putative transcription factors.

**In situ hybridization, SEM**. In situ hybridization was performed exactly as described in[4] with a minor change in the fixation protocol: here, we fixed the embryos for 1 h in 4% PFA/PBS at room temperature and washed the embryos several times first in PTw (1× PBS, 0.1% Tween 20) and then in 100% methanol prior to storing them at −20 °C. For the single chromogenic in situ hybridization, the RNA probes were detected with anti-Digoxigenin-AP Fab fragments (Roche) diluted 1:4000 in 0.5% blocking reagent (Roche) in 1× MAB followed by a substrate reaction with a mixture of NBT/BCIP as in[4]. Imaging was performed with a Nikon 80i compound microscope equipped with the Nikon DS-Fi1 camera. For the fluorescent double in situ hybridization, the hybridization protocol was similar to the single chromogenic in situ protocol except for the changes outlined below. FITC- and Dig-labeled RNA probes were simultaneously added to the sample. After stringent post-hybridization washes, the embryos were blocked in the 0.5% TSA Blocking Reagent (Perkin-Elmer) in TNT buffer for 1 h, and stained overnight at 4 °C with anti-Digoxigenin-POD Fab fragments (Roche) diluted 1:100 in blocking buffer. The unbound antibody was then removed by 10×10 min TNT washes, and the fluorescent signal was developed using the TSA Plus Cyanine 3 System (Perkin-Elmer) according to the manufacturer's protocol. The staining was stopped by multiple TNT washes, and peroxidase was inactivated by a 20 min wash in 1% $H_2O_2$/TNT in the dark. After that, the embryos were washed several times with TNT, blocked as described above and stained with the anti-Fluorescein-POD Fab fragments (Roche) diluted 1:50 in blocking buffer. Fluorescent signal was then developed as described above using the TSA Plus Fluorescein System (Perkin-Elmer). After stopping the staining with multiple TNT washes the embryos were embedded in Vectashield (Vectorlabs) and imaged with the Leica SP8 CLSM. Preparation of the samples for the SEM was performed as described in[4]. Imaging was done using the JEOL IT 300 scanning electron microscope.

**Phalloidin and antibody staining**. For phalloidin staining of fibrillar actin and anti-acetylated tubulin staining of cilia, the embryos were fixed in 4% PFA/PTwTx (1× PBS, 0.1% Tween 20, 0.2% Triton X100) for 1 h at room temperature, washed 5 times with PTwTx, incubated in 100% acetone pre-cooled to -20 °C for 7 min on ice, and washed 3 more times with PTwTx. Then, the embryos were incubated for 2 h in blocking solution (95% v/v of 1% BSA/PTwTx and 5% v/v of heat-inactivated sheep serum). Blocked embryos were stained overnight at 4 °C in 0.4U of Alexa Fluor 488 Phalloidin (ThermoFisher) and 0.1 µl of mouse monoclonal anti-acetylated tubulin (Sigma) dissolved in 100 µl blocking solution. Unbound primary antibody and phalloidin were washed away by five 10 min PTwTx washes, and the embryos were stained for 2 h at room temperature in the dark in 0.4U of Alexa Fluor 488 Phalloidin and 0.1 µl of Alexa Fluor 568 rabbit anti-mouse IgG (Molecular Probes) dissolved in 100 µl blocking solution. After five more 10 min PTwTx washes, the embryos were gradually embedded in Vectashield (Vectorlabs) and imaged with the Leica SP8 CLSM.

**Reporting summary**. Further information on research design is available in the Nature Research Reporting Summary linked to this article.

## Data availability

All data needed to evaluate the conclusions in the paper are present in the paper or the supplementary materials. Raw RNA-seq reads have been deposited in the NCBI BioProject database under the accession code: PRJNA661731.

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

## Acknowledgements

This work was funded by the Austrian Science Foundation (FWF) grant P30404-B29 to G.G., A.J.A. was supported by an HFSP postdoctoral fellowship (LT000809/2012-L). Y.K. was supported by the Lise Meitner FWF fellowship (M1140-B17). We are grateful to the Core Facility for Cell Imaging and Ultrastructure Research of the University of Vienna for the access to the confocal and scanning electron microscopes, Patrick Steinmetz for the fluorescent ISH protocol, Patricio Ferrer Murguia for an shRNA against *FoxA*, Saskia Hartmann for the initial genotyping of the *APC* mutants, Rohit Dnyansagar for the help with bioinformatics, Paul Knabl for drawing Fig. 7a, Emmanuel Haillot for discussions, and David Mörsdorf for critically reading the manuscript.

## Author contributions

T.L. performed the majority of the experiments, planned experiments and analyzed data; A.J.A and U.T. conceived the generation of the *APC* mutant line; A.J.A. generated the *APC* mutant line, and started its characterization together with A.D.; T.G. and I.N. performed treatments, and prepared RNA for sequencing; B.Z. supervised the bioinformatic analysis; Y.K. performed transplantations on *Bra* morphants; M.S. generated mosaic *EF1α::β-cat_stab* polyps; G.G. conceived the study, planned experiments, performed experiments, analyzed data and wrote the paper. All authors edited the paper.

## Competing interests

The authors declare no competing interests.
