## [Peer Review File · Nature Communications]

Reviewers' Comments:

Reviewer #1:

Remarks to the Author:

In this manuscript, Bagaeva et al., identified key transcriptional factors (TFs) controlling axial patterning in the *Nematostella* embryos. Beta-catenin signaling is a central patterning regulator of the main body axis across many animal species. To elucidate the regulatory logic of this well-conserved signaling in a non-bilaterian animal, the authors performed a series of experiments in the cnidarian *Nematostella*, belonging to the sister group of Bilateria. Using RNA-seq experiments in the context of pharmacological inhibition of GSK3Beta or APC mutant background, the authors generated a list of differentially expressed genes and focused on candidate TFs. Based on functional perturbations (shRNAs and morpholinos) and changes in the patterns of distinct transcriptional domains along the oral-aboral axis, the authors showed that more orally expressed TFs downstream of Beta-catenin signaling repress more aborally expressed beta-catenin targets, progressively restricting the initially aboral identity of the embryo. Given the similarity of this axial gene regulatory network to the A-P patterning in the sea urchin, the authors proposed a common evolutionary origin of beta-catenin-dependent embryonic patterning in cnidaria and Bilateria.

The manuscript is well-written, and the data is convincing. In general, this work would be of great interest to the evo-devo community. However, it would need some clarifications regarding the following points:

- 1- The current scoring of the in situ data does not reflect the variability of expression patterns and their corresponding domain changes under functional perturbations. As the conversion of this qualitative in situ images into quantitative expression domains is possible (see Botman and Kaandorp, 2012), the authors should take advantage of their large in situ data to provide a quantitative description of the patterning domains along the oral-aboral axis.
- 2- The plots in Fig1c are not supported by any experimental data. The authors have not shown the distribution and levels of Beta-catenin protein in the APC mutant or pharmacological inhibition of GSK3Beta.
- 3- The authors do not mention any information about the developmental phenotypes of the KDs of these transcriptional factors beyond the molecular changes.
- 4- The APC mutant has been previously described in Pukhlyakova et al., 2018. It is not clear that this is a novel mutant line that was generated in this study.

Minor comment:

- 5- All images are lacking scale bars

Reviewer #2:

Remarks to the Author:

In the manuscript by Bagaeva et al the authors report on the identification of genes required for the patterning of the oral-aboral axis in *Nematostella*. With an RNA-seq approach the authors identify the transcription factors Bra, FoxA, Lmx and FoxB and demonstrate by well-designed knockdown and in situ hybridization experiments that these transcription factors prevent the oral expansion of more aborally expressed beta-catenin targets. In addition, the authors identify Sp6-9 that is expressed in the midbody domain and prevents the oral expansion of the aboral domain. As a similar gene regulatory network is active during the patterning of sea urchins and vertebrates,

the authors propose that the cnidarian oral-aboral and the deuterostome posterior-anterior axes are equivalent.

The presented results are novel and have broad importance. However, before reaching a final decision the authors should address the comments below.

Do adult *Nematostella* require Bra, Lmx, FoxA and FoxB for the patterning of the oral-aboral axis or is the described gene regulatory network only active in embryos?

Page 6: "...*Nematostella* Sp5 is activated by beta-catenin signaling." Several studies showed that Sp5 is an important regulator for patterning in *Hydra* and vertebrates. What is the function of *Nematostella* Sp5? Does it act as a transcriptional repressor on any of the described genes?

Page 5: "We conclude that oral "window" genes are activated by beta-catenin signaling (either directly or indirectly), and repressed by beta-catenin dependent "saturating" transcription factors". Do Bra, Lmx, FoxA and FoxB contain TCF binding sites in their promoters?

Supplementary Figure 1h: the authors do not show in situ hybridizations for Axin, Wnt2, Wnt3 and WntA in wild-type animals, why?

Figure 1c: It is not clear what the blue and purple colors in the graph mean. It is also a bit confusing that beta-catenin and the repressor X have almost the same color. Furthermore, to help readers non-familiar with *Nematostella* please highlight the oral/aboral domains.

Figure 6a: Why do the authors indicate a repressing effect of Bra on its own expression when they show in Supplementary Figure 3a that the expression of Bra is reduced after its silencing? A model should be added showing only the experimentally identified interactions.

Page 2: "...we focus on the deciphering the mechanisms..." should read "...we focus on deciphering the mechanisms..."

Page 3: "RNASeq-based" should read "RNA-seq-based".

Supplementary Figure 1b, c: Please add arrows to highlight ectopic oral structures.

Supplementary discussion should be moved to the main discussion. Please better discuss how similar the *Nematostella* oral-aboral and the deuterostome posterior-anterior axes are.

All sequencing datasets should be made publically available.

Reviewer #3:

Remarks to the Author:

The manuscript Ancestral regulatory logic of the β -catenin dependent axial patterning, further investigates the role of β -catenin patterning of the cnidarian O/A axis in cnidarians. Through a logical series of experiments, the authors present a comprehensive and rigorous series of experiments, and carry out a high-quality analysis of the knockdowns and GSK3 β antagonist treatments. The data certainly add to our understanding of the patterning of O/A axis and is a worthy contribution to the literature on this topic that would be broadly interesting.

The first experimental sections identifying repressor X is very convincing, especially the progression from RNAseq to the short list of gene candidates, and how they exclude the different candidates to end up with the finale list of 4 genes.

The section of the identification of repressor Y would have benefitted from following the logic for the identification of repressor X. It is not clear how 26 gene candidates are whittled down to shortlist of three sp genes without explanation. The authors state "we looked at the transcription

factor coding genes downregulated by all treatments in our RNASeq experiment, and we found 25 such genes (Fig.4a). Two of them attracted our attention as members of the Sp family of the Krüppel-like transcription factors". What is the rationale for not considering the other candidates? The experiments are aimed at understanding how O/A patterning of the ectoderm occurs in cnidarians, but also to strengthen comparisons to A/V and A/P patterning bilaterians, particularly in deuterostomes. The first major manuscript on the role of B-cat in *Nematostella* in 2003 showed the onset of nuclearization of B-catenin at early blastula, right when they begin the Azk experiments in this manuscript. The Wikramanayake paper also showed a great expansion of what they then called "entoderm" following LiCl experiments along with the modification of O/A axis. These seem like discrete roles similar to what has been described in deuterostomes. Work in urchins, asteroids and hemichordates has described essentially two quite different phases of B-catenin patterning of early embryos – establishment of the vegetal pole and endomesoderm, then patterning of the A/P axis. Up until early/mid blastula embryos respond to AZK by becoming vegetalized by expansion of genes like FoxA that define the endoderm. However, by mid-blastula embryos no longer expand endomesodermal markers in response to increase of β -catenin stabilization, and A/P patterning begins with the onset of Wnt zygotic expression. There is likely some temporal overlap in both endomesoderm specification and ectodermal patterning, but they can be largely separated experimentally. The vegetal pole is first established, and then this is the source of posteriorizing factors that subsequently provide pattern to the ectoderm. The way the data are presented in this manuscript, the authors seem to imply that establishment of the endoderm, patterning of the ectoderm along the O/A axis represent a single continuous role, rather than discrete roles for B-catenin. The AZK treatments begin at early blastula, right at the onset of B-cat stabilization and continues to mid gastrula, which may confound two separate roles of B-cat. Azk treatments expand the endodermal and oral ectodermal repressor genes throughout the embryo. My question is when ectodermal patterning begins, and whether it is after the establishment of the endodermal territory, similar to that found in invertebrate deuterostomes? If ectodermal patterning happens later, with the onset of zygotic gene expression, then I have a hard time with some of the interpretations of the experiments. Clearly the repressors they have described play an important role in the establishment of the most oral territory and delineating the boundary between endoderm and ectoderm, but I question their relevance for understanding patterning most of the ectodermal territory. Driving expression of these oral repressors into the ectoderm by early Azk treatments may be confounding early and late roles of β -catenin. How would interpretations be different if the AZK treatments were delayed until the completion of endoderm specification? Do oral ectodermal markers like ectodermal Brachyury initiate early, coincident with the nuclear β -catenin in the oral side? When does ectodermal patterning begin? My perspective is biased towards an understanding of bilaterians early embryos, but I would appreciate some clarification of this as close comparisons are being drawn to mechanisms of bilaterian A/V and A/P patterning, which I understand as being pretty discrete patterning events. I'm happy to be corrected if I have misunderstood how this process works in *Nematostella*. Despite the great experiments, I feel the manuscript could be greatly improved with a more comprehensive introduction and discussion to more fully introduce the topic, including the history of the work on this question. I still don't fully understand the idea of a "window" gene, which they can clear up quickly with a better description. Similarly, the discussion is very perfunctory and misses an opportunity to put their data into a broader phylogenetic context that the title promises. The authors conclude "We conclude that these processes share a common evolutionary origin predating the cnidarian-bilaterian split, and propose that cnidarian oral-aboral and the deuterostome posterior-anterior body axes are equivalent". This is a strong conclusion but alternatives are not discussed – I tend to agree with their interpretation. In summary, the data presented are compelling, very comprehensive, and of high quality and extend our mechanistic knowledge of O/A patterning in cnidarians. The manuscript would be greatly improved with a more informative introduction and discussion to put their work into the context of previous work and alternative interpretations. Additionally, my major concerns revolve around the timing of their experiments using AZK and whether this results in confounding two quite separate roles of β -catenin in endoderm specification and later ectodermal patterning. This

would need to be clarified before I could recommend publication.

POINT-BY-POINT RESPONSES TO THE REVIEWER COMMENTS (IN RED)

Reviewer #1 (Remarks to the Author):

In this manuscript, Bagaeva et al., identified key transcriptional factors (TFs) controlling axial patterning in the *Nematostella* embryos. Beta-catenin signaling is a central patterning regulator of the main body axis across many animal species. To elucidate the regulatory logic of this well-conserved signaling in a non-bilaterian animal, the authors performed a series of experiments in the cnidarian *Nematostella*, belonging to the sister group of Bilateria. Using RNA-seq experiments in the context of pharmacological inhibition of GSK3Beta or APC mutant background, the authors generated a list of differentially expressed genes and focused on candidate TFs. Based on functional perturbations (shRNAs and morpholinos) and changes in the patterns of distinct transcriptional domains along the oral-aboral axis, the authors showed that more orally expressed TFs downstream of Beta-catenin signaling repress more aborally expressed beta-catenin targets, progressively restricting the initially aboral identity of the embryo. Given the similarity of this axial gene regulatory network to the A-P patterning in the sea urchin, the authors proposed a common evolutionary origin of beta-catenin-dependent embryonic patterning in cnidaria and Bilateria.

The manuscript is well-written, and the data is convincing. In general, this work would be of great interest to the evo-devo community. However, it would need some clarifications regarding the following points:

1- The current scoring of the in situ data does not reflect the variability of expression patterns and their corresponding domain changes under functional perturbations. As the conversion of this qualitative in situ images into quantitative expression domains is possible (see Botman and Kaandorp, 2012), the authors should take advantage of their large in situ data to provide a quantitative description of the patterning domains along the oral-aboral axis.

We do not think that expression variability analysis is necessary for this paper. The knockdown phenotypes presented in the paper are extremely robust and clear – these are not slight shifts of expression domains but major changes in the expression, and their penetrance is usually very high – please see quantifications on each figure demonstrating the proportion of the embryos

displaying the same phenotype as shown on the photo. Also, for each gene we knocked down, at least two independent means of knockdown have been used, and the effects on downstream genes were confirmed to be near-identical (see Fig. 3 and Suppl. Fig. 4, also see Suppl. Fig. 11).

Finally, the paper describes over 400 different perturbation/analyzed gene combinations, and the phenotypes consistently match and support each other. Quantifying the variability of the expression as in Botman et al., 2015 for 428 perturbation/analyzed gene combinations where the phenotype is really clear will represent an enormous amount of work without a potential for changing any of the conclusions of our paper. Upon request, we are, of course, happy to photograph and provide images of all embryos from any perturbation/analyzed gene of Reviewer's choice, so that the Reviewer could convince him/herself that the embryos shown in the paper are representative.

2- The plots in Fig 1c are not supported by any experimental data. The authors have not shown the distribution and levels of Beta-catenin protein in the APC mutant or pharmacological inhibition of GSK3Beta.

That is correct, thanks for pointing that out. Unfortunately, multiple tests by us and others showed that there is currently no antibody available, which would permit the quantification of the distribution of zygotic nuclear beta-catenin in the *Nematostella* embryos. Fig. 1c is a hypothetical model providing a plausible explanation for the “window” and “saturating” behavior of genes in response to the azakenpaullone treatment. We corrected the figure legend of the Fig. 1c to make clear that it represents our working hypothesis.

3- The authors do not mention any information about the developmental phenotypes of the KDs of these transcriptional factors beyond the molecular changes.

There is no observable morphological change in the development of the KD embryos by gastrula stage. We include a description of the later development of the *Bra*, *FoxA*, *FoxB*, *Lmx*, and *Sp6-9* KD embryos in the revised version of the paper (See Supplementary Results and Discussion 3 and Supplementary Fig. 9).

4- The APC mutant has been previously described in Pukhlyakova et al., 2018. It is not clear that this is a novel mutant line that was generated in this study.

The *APC* mutant line is the same line used by Pukhlyakova et al., 2018. However, there was no

explanation in that paper regarding the generation of the line and no characterization of the mutants beyond the analysis of *Brachyury* expression. We corrected the text and added the reference to Pukhlyakova et al. in our paper. We thank the Reviewer for pointing this out.

Minor comment:

5- All images are lacking scale bars

We added the scale bars to all figures

Reviewer #2 (Remarks to the Author):

In the manuscript by Bagaeva et al the authors report on the identification of genes required for the patterning of the oral-aboral axis in *Nematostella*. With an RNA-seq approach the authors identify the transcription factors Bra, FoxA, Lmx and FoxB and demonstrate by well-designed knockdown and in situ hybridization experiments that these transcription factors prevent the oral expansion of more aborally expressed beta-catenin targets. In addition, the authors identify Sp6-9 that is expressed in the midbody domain and prevents the oral expansion of the aboral domain. As a similar gene regulatory network is active during the patterning of sea urchins and vertebrates, the authors propose that the cnidarian oral-aboral and the deuterostome posterior-anterior axes are equivalent.

The presented results are novel and have broad importance. However, before reaching a final decision the authors should address the comments below.

Do adult *Nematostella* require Bra, Lmx, FoxA and FoxB for the patterning of the oral-aboral axis or is the described gene regulatory network only active in embryos?

Unfortunately, this interesting question is currently impossible to answer. In contrast to the situation in *Hydra*, where axial patterning is occurring constantly in adult polyps due to the tissue dynamics, it is not clear whether this is the case in *Nematostella*. Also, currently, we do not have the means to knock genes down in adult *Nematostella*.

Page 6: "...*Nematostella* Sp5 is activated by beta-catenin signaling." Several studies showed that Sp5 is an important regulator for patterning in *Hydra* and vertebrates. What is the function of *Nematostella* Sp5? Does it act as a transcriptional repressor on any of the described genes?

We agree that understanding whether *Nematostella* Sp5 is a β -catenin-dependent antagonist of β -catenin signaling, as shown in *Hydra*, is an extremely interesting question. However, we followed the advice of the Reviewer 3 and restructured the "repressor Y responsible for the midbody" part of the paper by adopting the same candidate elimination strategy we used when searching for the "repressor X responsible for the oral domain". Since *Sp5* fulfills neither the "repressor X candidate" criterion of being upregulated in all our treatments, nor the "repressor Y candidate" criterion of being repressed in all our treatments, we removed *Sp5* from this paper and will address its function in the future studies.

Page 5: "We conclude that oral "window" genes are activated by beta-catenin signaling (either directly or indirectly), and repressed by beta-catenin dependent "saturating" transcription factors". Do *Bra*, *Lmx*, *FoxA* and *FoxB* contain TCF binding sites in their promoters?

Currently, we do not have ChIP-grade antibodies for *Nematostella* Tcf, so we can only rely on the bioinformatic prediction of the Tcf binding sites, which may or may not be correct. According to PROMO (http://alggen.lsi.upc.es/cgi-bin/promo_v3/promo/promoinit.cgi?dirDB=TF_8.3), both the "saturating" genes *Bra*, *FoxB*, *FoxA*, and *Lmx*, as well as the "window" genes *Wnt1* and *Wnt2* have multiple TCF sites in their putative promoters. However, although this fits our hypothesis, we do not feel comfortable using this as an argument in favor of the "transcriptional repressor X" hypothesis in our paper without experimental proof, especially since the regulation does not necessarily have to be direct (as we mention in the text).

Supplementary Figure 1h: the authors do not show in situ hybridizations for Axin, Wnt2, Wnt3 and WntA in wild-type animals, why?

The wild type expression of all these four genes at the gastrula stage has been published previously (e.g. in Kraus et al., 2016, DOI: 10.1038/ncomms11694 and elsewhere). Here, they are used as markers to describe the effect of the *APC* mutation. All embryos on Suppl. Fig. 1h are siblings from the same cross of male and female heterozygous *APC* mutants, which have been fixed and hybridized to different probes. Among these embryos, 25% are expected to be

homozygous mutants, 50% - heterozygous mutants, and 25% - wild types. Individual embryos with unknown genotypes were imaged, then their genomic DNA was extracted, the mutation site was amplified by PCR, and, finally, these imaged individual embryos were genotyped by Sanger sequencing. Since the expression of *Axin*, *Wnt2*, *Wnt3* and *WntA* in *APC* heterozygous animals does not differ from the published wild type expression, we took representative images of embryos with known genotypes independent of whether they were wild type or heterozygous.

Figure 1c: It is not clear what the blue and purple colors in the graph mean. It is also a bit confusing that beta-catenin and the repressor X have almost the same color. Furthermore, to help readers non-familiar with *Nematostella* please highlight the oral/aboral domains.

We revised Fig. 1c and also significantly expanded the figure legend to make it clear. The oral-aboral axis is now marked on the graphs of the Fig. 1c.

Figure 6a: Why do the authors indicate a repressing effect of Bra on its own expression when they show in Supplementary Figure 3a that the expression of Bra is reduced after its silencing? A model should be added showing only the experimentally identified interactions.

We know that *Brachyury* is repressing its own expression because we knocked it down by two independent experimental means. When *Brachyury* is knocked down by RNAi, its expression is reduced upon knockdown (as on Suppl. Fig. 3a and Suppl. Fig. 5), however, when *Brachyury* is knocked down by a translation-blocking morpholino, *Brachyury* mRNA expression is upregulated (as on Suppl. Fig. 4b), while the effect of the morpholino-mediated knockdown on the downstream *Wnt* genes is the same as in the RNAi. The reason for that is most likely that the lack of *Brachyury* protein de-represses *Brachyury* transcription, which is detectable in the morphants, but not in the RNAi embryos, since the overproduced *Brachyury* mRNA is also destroyed by RNAi. The negative regulation of *Bra* expression upon MO-mediated *Bra* knockdown is discussed in the Supplementary Results and Discussion 1.

Page 2: "...we focus on the deciphering the mechanisms..." should read "...we focus on deciphering the mechanisms..."

We corrected this

Page 3: “RNASeq-based” should read “RNA-seq-based”.

We corrected this

Supplementary Figure 1b, c: Please add arrows to highlight ectopic oral structures.

Done

Supplementary discussion should be moved to the main discussion.

We considered this option, but eventually decided to leave it in the supplement. In our opinion, the discussion of the topology of the network, the transplantation experiments and the newly added description of the later development phenotypes (upon request by the Reviewer 1) makes the main text too long, much more difficult to read and distracts the attention of the reader from the main conclusions regarding the general logic of the O-A patterning.

Please better discuss how similar the *Nematostella* oral-aboral and the deuterostome posterior-anterior axes are.

We have significantly expanded the introduction and the discussion of the paper with the focus on this point.

All sequencing datasets should be made publically available.

All sequences are freely available at NCBI (BioProject PRJNA661731)

Reviewer #3 (Remarks to the Author):

The manuscript Ancestral regulatory logic of the β -catenin dependent axial patterning, further investigates the role of β -catenin patterning of the cnidarian O/A axis in cnidarians. Through a logical series of experiments, the authors present a comprehensive and rigorous series of experiments, and carry out a high-quality analysis of the knockdowns and GSK3 β antagonist treatments. The data certainly add to our understanding of the patterning of O/A axis and is a

worthy contribution to the literature on this topic that would be broadly interesting.

The first experimental sections identifying repressor X is very convincing, especially the progression from RNAseq to the short list of gene candidates, and how they exclude the different candidates to end up with the finale list of 4 genes.

The section of the identification of repressor Y would have benefitted from following the logic for the identification of repressor X. It is not clear how 26 gene candidates are whittled down to shortlist of three sp genes without explanation. The authors state “we looked at the transcription factor coding genes downregulated by all treatments in our RNASeq experiment, and we found 25 such genes (Fig.4a). Two of them attracted our attention as members of the Sp family of the Krüppel-like transcription factors”. What is the rational for not considering the other candidates?

We agree with the Reviewer, and we did consider other candidates. We used the same search logic for repressor Y as we did for the repressor X. Since by now we succeeded in knocking down all the necessary repressor Y candidates with two independent shRNAs, we have completely re-written this part and show the similar logic of searching for the repressor Y as in the first part of the manuscript. The expression analysis of all 25 genes and the knockdowns of all the candidates other than Sp6-9 are presented on the new Supplementary Figs. 10-11.

The experiments are aimed at understanding how O/A patterning of the ectoderm occurs in cnidarians, but also to strengthen comparisons to A/V and A/P patterning bilaterians, particularly in deuterostomes. The first major manuscript on the role of B-cat in *Nematostella* in 2003 showed the onset of nuclearization of B-catenin at early blastula, right when they begin the Azk experiments in this manuscript. The Wikramanayake paper also showed a great expansion of what they then called “entoderm” following LiCl experiments along with the modification of O/A axis. These seem like discrete roles similar to what has been described in deuterostomes. Work in urchins, asteroids and hemichordates has described essentially two quite different phases of B-catenin patterning of early embryos – establishment of the vegetal pole and endomesoderm, then patterning of the A/P axis. Up until early/mid blastula embryos respond to AZK by becoming vegetalized by expansion of genes like FoxA that define the endoderm. However, by mid-blastula embryos no longer expand endomesodermal markers in response to increase of β -catenin stabilization, and A/P patterning begins with the onset of Wnt zygotic expression. There is likely some temporal overlap in both endomesoderm

specification and ectodermal patterning, but they can be largely separated experimentally. The vegetal pole is first established, and then this is the source of posteriorizing factors that subsequently provide pattern to the ectoderm. The way the data are presented in this manuscript, the authors seem to imply that establishment of the endoderm, patterning of the ectoderm along the O/A axis represent a single continuous role, rather than discrete roles for B-catenin.

Endoderm specification clearly precedes ectodermal patterning in *Nematostella*. We discuss the literature and add experimental data showing this (Fig. 6). Please also see the comment below.

The AZK treatments begin at early blastula, right at the onset of B-cat stabilization and continues to mid gastrula, which may confound two separate roles of B-cat. Azk treatments expand the endodermal and oral ectodermal repressor genes throughout the embryo. My question is when ectodermal patterning begins, and whether it is after the establishment of the endodermal territory, similar to that found in invertebrate deuterostomes? If ectodermal patterning happens later, with the onset of zygotic gene expression, then I have a hard time with some of the interpretations of the experiments. Clearly the repressors they have described play an important role in the establishment of the most oral territory and delineating the boundary between endoderm and ectoderm, but I question their relevance for understanding patterning most of the ectodermal territory. Driving expression of these oral repressors into the ectoderm by early AzK treatments may be confounding early and late roles of β -catenin. How would interpretations be different if the AZK treatments were delayed until the completion of endoderm specification? Do oral ectodermal markers like ectodermal Brachyury initiate early, coincident with the nuclear β -catenin in the oral side? When does ectodermal patterning begin? My perspective is biased towards an understanding of bilaterians early embryos, but I would appreciate some clarification of this as close comparisons are being drawn to mechanisms of bilaterian A/V and A/P patterning, which I understand as being pretty discrete patterning events. I'm happy to be corrected if I have misunderstood how this process works in *Nematostella*.

We thank the Reviewer for bringing up these extremely valid points. First of all, we apologize for a mistake we made in the Methods and in the legend to Fig. 1e: our AZK incubations started not at 6 hpf, but at 10 hpf, and lasted until 30 hpf and not 24 hpf. This mistake is now corrected.

Thus, all the AZK treatments we showed in the first version of the manuscript started after the documented accumulation of β -catenin on one side of the embryo shown in Wikramanayake et al., 2003 and in Kraus et al., 2016. Importantly, in all our experiments, gastrulation was not affected by AZK treatment starting at 10 hpf. In contrast, Leclere et al., 2016 showed and we here confirmed that the treatment with the same AZK concentration starting at 2 hpf (Leclere et al., 2016) or 3 hpf (our study) blocks endoderm specification. Therefore, ectodermal patterning clearly starts after endoderm specification in *Nematostella*, like in Bilateria.

Importantly, *Brachyury*, *FoxA* and *FoxB* are never expressed in the *Nematostella* endoderm – they are markers of the oral and later also pharyngeal ectoderm, which is a derivative of the invaginated blastopore lip. Notably, based on a large set of genetic markers, Steinmetz et al., 2017 proposed that cnidarian pharyngeal ectoderm and endoderm may be homologous, respectively, to the bilaterian endoderm and mesoderm. If true, this hypothesis makes *Nematostella Bra* and *FoxA* “endodermal” rather than ectodermal markers, restoring the situation similar to bilaterian development. However, in addition to the expression in the pharynx, which may be interpreted as “endoderm” according to Steinmetz et al., 2017, all four repressor X genes are also expressed in the body wall ectoderm on the oral end of the embryo, and not only in the pharynx. This is different to the situation in ambulacrarians, where *FoxA* is an endodermal marker and does seem to be expressed in the posterior ectoderm.

In the revised version of the manuscript, we include new experiments with AZK treatments showing that endoderm specification precedes ectodermal patterning, and that “window” and “saturating” behavior in response to AZK does not depend on the presence or absence of the endodermal territory (Fig. 6). This result is additionally addressed in the Discussion.

Despite the great experiments, I feel the manuscript could be greatly improved with a more comprehensive introduction and discussion to more fully introduce the topic, including the history of the work on this question. I still don't fully understand the idea of a “window” gene, which they can clear up quickly with a better description. Similarly, the discussion is very perfunctory and misses an opportunity to put their data into a broader phylogenetic context that the title promises. The authors conclude “We conclude that these processes share a common evolutionary origin predating the cnidarian-bilaterian split, and propose that cnidarian oral-aboral and the deuterostome posterior-anterior body axes are equivalent”. This is a strong conclusion but alternatives are not discussed – I tend to agree with their interpretation.

In summary, the data presented are compelling, very comprehensive, and of high quality and extend our mechanistic knowledge of O/A patterning in cnidarians. The manuscript would be greatly improved with a more informative introduction and discussion to put their work into the context of previous work and alternative interpretations. Additionally, my major concerns revolve around the timing of their experiments using AZK and whether this results in confounding two quite separate roles of β -catenin in endoderm specification and later ectodermal patterning. This would need to be clarified before I could recommend publication.

We have re-worked the introduction and the discussion to cover the questions raised by the Reviewer. We thank him/her for the extremely valuable comments.

Reviewers' Comments:

Reviewer #1:

Remarks to the Author:

I am satisfied by the changes that the authors have made to the revised MS. For the scoring of gene expression, the authors should then indicate in the figure legend the meaning of the numbers associated with the in situ images.

Reviewer #2:

Remarks to the Author:

The authors have satisfactorily revised their manuscript that can now be accepted for publication

Reviewer #3:

Remarks to the Author:

I am satisfied with the responses to my comments and feel the manuscript is greatly improved.